# Fraxinol Stimulates Melanogenesis in B16F10 Mouse Melanoma Cells through CREB/MITF Signaling

**DOI:** 10.3390/molecules27051549

**Published:** 2022-02-25

**Authors:** Sun Young Moon, Kazi-Marjahan Akter, Mi-Jeong Ahn, Kwang Dong Kim, Jiyun Yoo, Joon-Hee Lee, Jeong-Hyung Lee, Cheol Hwangbo

**Affiliations:** 1Division of Applied Life Science (BK21), PMBBRC and Research Institute of Life Sciences, Gyeongsang National University, Jinju 52828, Korea; symoon0414@gnu.ac.kr (S.Y.M.); kdkim88@gnu.ac.kr (K.D.K.); yooj@gnu.ac.kr (J.Y.); 2Division of Life Science, College of Natural Sciences, Gyeongsang National University, Jinju 52828, Korea; 3College of Pharmacy and Research Institute of Pharmaceutical Sciences, Gyeongsang National University, Jinju 52828, Korea; marjahan7silva@gmail.com (K.-M.A.); amj5812@gnu.ac.kr (M.-J.A.); 4Department of Animal Bioscience, College of Agriculture and Life Sciences, Gyeongsang National University, Jinju 52828, Korea; sbxjhl@gnu.ac.kr; 5Department of Biochemistry (BK21), College of Natural Sciences, Kangwon National University, Chuncheon 24341, Korea; jhlee36@kangwon.ac.kr

**Keywords:** fraxinol, B16-F10 cells, melanogenesis, microphthalmia-associated transcription factor, depigmentation

## Abstract

Melanin pigment produced in melanocytes plays a protective role against ultraviolet radiation. Selective destruction of melanocytes causes chronic depigmentation conditions such as vitiligo, for which there are very few specific medical treatments. Here, we found that fraxinol, a natural coumarin from *Fraxinus* plants, effectively stimulated melanogenesis. Treatment of B16-F10 cells with fraxinol increased the melanin content and tyrosinase activity in a concentration-dependent manner without causing cytotoxicity. Additionally, fraxinol enhanced the mRNA expression of melanogenic enzymes such as tyrosinase, tyrosinase-related protein-1, and tyrosinase-related protein-2. Fraxinol also increased the expression of microphthalmia-associated transcription factor at both mRNA and protein levels. Fraxinol upregulated the phosphorylation of cyclic adenosine monophosphate (cAMP) response element-binding protein (CREB). Furthermore, H89, a cAMP–dependent protein kinase A inhibitor, decreased fraxinol-induced CREB phosphorylation and microphthalmia-associated transcription factor expression and significantly attenuated the fraxinol-induced melanin content and intracellular tyrosinase activity. These results suggest that fraxinol enhances melanogenesis via a protein kinase A-mediated mechanism, which may be useful for developing potent melanogenesis stimulators.

## 1. Introduction

Melanogenesis is a complex biological and biochemical process involving melanin synthesis [1]. Melanin pigment is produced in melanosomes, which are specialized organelles, and melanocytes, which are neural crest-derived dendritic cells distributed among keratinocytes in the basal layer of the epidermis [2]. Melanocytes are involved in transferring melanosomes containing melanin to the surrounding keratinocytes, resulting in the deposition of melanin in the epidermis of the human skin [3]. The proper biological functioning of these cells is essential for normal pigmentation. Hypo- or hyperpigmentation disorders are caused by alterations in the number of melanocytes or dysregulation of melanin synthesis in melanocytes [4].

Vitiligo is one of the most common acquired hypopigmented skin disorders, affecting approximately 0.5–2% of individuals worldwide. The incidence rate does not differ by race, skin type, or sex [5]. A recent study reported that multiple pathogenic mechanisms are involved in vitiligo, including genetic changes, oxidative stress, and abnormal innate and adaptive immunities, which lead to melanocyte destruction [6]. This heterogeneity makes the management of vitiligo difficult. Thus, agents that stimulate melanogenesis have been widely investigated for repigmentation [7].

Mammalian melanin synthesis is directly catalyzed by melanogenic enzymes, including tyrosinase (TYR), TYR-related protein-1 (TRP1), and TYR-related protein-2 (TRP2) [8]. TYR is a central enzyme in the melanin pigment formation pathway and is involved in the hydroxylation of tyrosine to 3,4-dihydroxyphenylalanine (DOPA) and oxidation of DOPA to DOPA quinone [9]. Other proteins (TRP1 and TRP2) are essential for the regulatory step that follows the action of TYR [10,11]. Microphthalmia-associated transcription factor (MITF) is a pivotal transcription factor that regulates both cardinal and melanogenic genes, including three melanocyte-specific genes (*TYR*, *TRP1*, and *TRP2*) which regulate the survival, proliferation, differentiation, and pigmentation of melanocytes [12]. Therefore, MITF plays an essential role in regulating melanin pigment levels. Many studies have shown that various molecular signaling pathways regulate melanogenesis [13]. Recent studies revealed that cyclic adenosine monophosphate (cAMP)/cAMP-dependent protein kinase A (PKA) is a major intracellular signaling pathway that activates the phosphorylation of cAMP response element-binding protein (CREB), leading to transcriptional activation of MITF [14]. Furthermore, activation of melanogenic proteins promotes MITF expression.

Although several management strategies are currently available to stop depigmentation and stimulate repigmentation, effective medical treatment remains a major challenge for patients with vitiligo [15]. Therefore, medicinal plant compounds have been considered for treating pigmentary diseases, including vitiligo, with negligible long-term side effects [16,17].

Melanogenesis stimulators have been synthesized and developed from plant extracts and active natural products [18]. Some studies showed that some natural compounds have strong melanogenesis-stimulating effects and suggest that they may be a source for developing anti-vitiligo agents [19,20]. *Fraxinus* species have been used as traditional medicines because of their various biological functions, such as anticancer, anti-inflammatory, antioxidant, neuroprotective, antifungal, and anti-aging activities [21,22]. Fraxinol is a hydroxycoumarin typically present in *Fraxinus* species [23,24]. Recent studies have reported that fraxinol has anti-lipase activity, but does not show an inhibitory effect against *Helicobacter pylori* [25,26]. However, there are no reports on the melanogenesis-related biological activity of fraxinol. In this study, we investigated the effect of fraxinol on melanogenesis and its underlying molecular mechanisms in B16-F10 cells.

## 2. Results

### 2.1. Fraxinol Induces the Production of Cellular Melanin in B16-F10 Melanoma Cells without Cytotoxicity

We investigated the melanogenesis-inducing effect of fraxinol (Figure 1A) in mouse melanoma B16-F10 cells. To examine the cytotoxic or proliferative effects of fraxinol on B16-F10 cells, cell viability was assessed using the 3-(4,5-dimethylthiazol-2-yl)-2,5-diphenyltetrazolium bromide (MTT) assay. B16-F10 cells were treated with fraxinol (0, 20, 40, 60, or 100 μM) for 48 h. As shown in Figure 1B, there were no significant changes in cell viability. Fraxinol-induced melanogenesis was assessed by measuring the accumulation of intracellular melanin in B16F10 cells. Fraxinol increased the cellular melanin content in a dose-dependent manner (Figure 1C). We also compared the fraxinol-induced melanin content with the alpha-melanocyte stimulating hormone (α-MSH)-stimulated melanin content (Figure 1D). Compared to untreated cells, fraxinol and α-MSH increased melanin levels by approximately 2.5- and 3-fold, respectively. These results confirmed that fraxinol induces melanogenesis activity in B16F10 cells without affecting cell survival and proliferation.

### 2.2. Fraxinol Increases Intracellular TYR Activity in B16-F10 Cells

TYR is the first rate-limiting enzyme in melanin biosynthesis among the three melanogenesis-controlling enzymes, TYR, TRP1, and TRP2 [27]. To investigate the fundamental mechanisms by which fraxinol increases the melanin content in B16-F10 cells, we examined its effect on TYR activity. We measured both intracellular and cell-free mushroom TYR activity using L-DOPA as a substrate. As shown in Figure 2A, fraxinol significantly and dose-dependently increased intracellular TYR activity. However, cell-free mushroom TYR-based enzyme activity was not changed after fraxinol treatment (Figure 2B), indicating that fraxinol has no direct effect on TYR activity in cell-free systems. Furthermore, we confirmed that fraxinol enhanced intracellular TYR activity by performing microscopic analysis after DOPA staining (Figure 2C). These results indicated that fraxinol increased melanin synthesis by upregulating TYR activity in B16-F10 cells. However, this increase in TYR activity did not occur through direct regulation.

### 2.3. Fraxinol Enhances the Expression Level of Melanogenic Enzymes

Three melanogenic enzymes (TYR, TRP1, and TRP2) are indispensable for melanin biosynthesis in melanocytes; however, the functions of TRP1 and TRP2 are not completely understood [27]. Therefore, we investigated the effect of fraxinol on the mRNA and protein expression levels of TYR, TRP1, and TRP2, considering that fraxinol increased intracellular TYR activity. The mRNA expression levels of TYR, TRP-1, and TRP-2 were detected using quantitative reverse transcription polymerase chain reaction (qRT-PCR). We found that fraxinol significantly increased the mRNA levels of all three enzymes in B16F10 cells (Figure 3A–C). Consistent with these results, fraxinol enhanced the protein expression of TRP-1 after treatment for 48 h (Figure 3D). However, the TYR and TRP2 protein expression levels did not increase significantly. These results suggest that TRP1 expression is involved in fraxinol-induced melanin synthesis.

### 2.4. Fraxinol Upregulates the Phosphorylation of CREB and Expression Level of MITF

Melanogenic enzymes are regulated by MITF, an essential transcription factor for melanin synthesis and melanocyte development [13]. MITF is regulated by multiple transcription factors, among which activated CREB positively regulates MITF gene expression by binding to the MITF promoter site in melanocytes [28]. To analyze the mechanism underlying the melanogenesis-inducing effects of fraxinol, we examined the mRNA and protein expression levels of MITF in B16-F10 cells. We found that fraxinol upregulated MITF expression at both the mRNA and protein levels (Figure 4A,B). These results suggest that fraxinol-induced increases in melanogenic enzymes is attributed to increased MITF expression. Next, we examined the phosphorylation levels of CREB. As shown in Figure 4C, the expression levels of phosphorylated CREB were enhanced by fraxinol treatment. These findings indicate that fraxinol activates melanogenesis by increasing MITF expression and phosphorylating CREB in B16-F10 cells.

### 2.5. PKA Inhibitor Attenuates CREB Phosphorylation and MITF Expression Induced by Fraxinol

CREB/MITF-mediated activation of melanogenesis is regulated by several melanogenic signaling pathways, including the PKA-mediated pathway [8,29]. Particularly, it has been well-established that agonist-bound melanocortin-1 receptor activates the PKA enzyme and phosphorylation of CREB, which induces MITF expression [2]. Therefore, to clarify the upstream molecular mechanism of fraxinol-induced CREB/MITF activation, we examined the effects of a PKA inhibitor (H89) on CREB phosphorylation and MITF expression. As shown in Figure 5A–C, suppression of the PKA pathway using H89 led to a decrease in fraxinol-induced CREB phosphorylation and MITF mRNA and protein expression. In addition, H89 decreased fraxinol-induced TRP1 expression (Figure 5D). These results suggest that PKA signaling is involved in fraxinol-induced melanogenesis.

### 2.6. PKA Inhibitor Reduces Melanin Production and the Increased Tyrosinase Activity by Fraxinol Treatment

We further explored the effects of the PKA inhibitor on the fraxinol-induced melanin content and TYR activity. As shown in Figure 6A–C, fraxinol treatment increased TYR activity and melanin content; these effects were remarkably diminished by the PKA inhibitor. These results indicate that PKA-mediated signaling is essential for melanin production and tyrosinase activity increased by fraxinol.

## 3. Discussion

Melanin synthesis is a crucial physiological process for protection against ultraviolet irradiation-induced damage, detoxification, and embryonic development [30,31]. Melanin pigmentation disorders include hypopigmentation due to decreased melanin production and hyperpigmentation due to increased melanin production [32]. Inhibitors that reduce the high production of melanin in hyperpigmentation have been more broadly studied than stimulators that induce melanin production in hypopigmentation from various sources (plant extracts, active natural products, and synthesized derivatives) [18,33]. More extensive studies are required to identify and develop new melanogenesis stimulators from natural sources for treating hypopigmentation diseases. Therefore, it is important to identify novel stimulators of melanogenesis.

Fraxinol is a natural compound present in *Fraxinus* species, commonly known as the ash tree [34]. However, the melanogenesis-related bioactivity of fraxinol has not been reported. We confirmed that fraxinol was not cytotoxic at concentrations of up to 100 μM. We used αMSH (100 nM) as a positive control to induce melanin production. Previous studies confirmed that α-MSH stimulates melanogenesis in melanocytes by activating the melanocortin 1 receptor, with the melanin-inducing effect saturated at approximately 100 nM [35,36,37,38]. Fraxinol and α-MSH increased the levels of cellular melanin by 2.5- and 3-fold compared to that in non-treated cells. Plants or active small molecules of natural origin have been reported to stimulate melanogenesis. Among the active compounds, ginsenoside and glycyrrhizin showed the strongest melanogenesis-inducing effects at concentrations of 100–300 μM and 1 mM, respectively, compared to fraxinol [39,40]. Fraxinol enhanced intracellular TYR activity in a dose-dependent manner but did not affect cell-free mushroom TYR activity. This result indicates that fraxinol does not directly affect TYR activity.

Fraxinol at a concentration of 100 μM was used for further mechanistic studies in B16F10 cells. Fraxinol significantly increased the mRNA and protein expression levels of all three melanogenic enzymes (TYR, TRP1, and TRP2) and protein expression of TRP1. However, TYR and TRP2 protein levels were not significantly increased by fraxinol treatment. Although TYR is a central enzyme involved in pigmentation, TRP1 and TRP2 play crucial roles in melanogenesis [10,11]. TRP1 catalyzes the oxidation of 5,6-dihydroxyindole-2-carboxylic acid to a carboxylated indole-quinone downstream of the melanin biosynthetic pathway [27]. TRP1 activity is involved in further metabolism of 5,6-dihydroxyindole-2-carboxylic acid to high-molecular weight pigmented biopolymers and is distinguished from TYR and TRP2 [41]. Furthermore, fraxinol upregulated the expression of MITF, a key transcription factor that regulates melanogenic enzyme transcription. MITF is modulated by multiple transcription factors, among which activated CREB positively regulates MITF expression [42,43]. The phosphorylation of CREB was enhanced at 4 h after fraxinol treatment and was still increased at 24 h (data not shown). This result indicates that increased phosphorylated CREB contributes to the enhancement of melanin production until 24 h after treatment with fraxinol. A PKA inhibitor attenuated CREB phosphorylation and MITF expression, thereby reducing TRP1 expression. We further confirmed that the PKA inhibitor dramatically reduced fraxinol-induced melanin synthesis and TYR activity. Based on these results, fraxinol may stimulate melanogenesis by activating PKA-dependent CREB/MITF/TRP1 signaling. Ginsenosides, also, stimulate melanogenesis via the PKA/CREB/MITF pathway [39] and glycyrrhizin induces melanogenesis by blocking glycogen synthase kinase 3β phosphorylation and by enhancing CREB phosphorylation [40].

Although different single or combination treatments have been broadly used against hypopigmentary disorders such as vitiligo, effective medical treatments are needed because patients either do not respond to treatment or develop undesirable side effects following long-term treatment [6,44]. Thus, the discovery and development of new melanogenic stimulators with fewer side effects are required. In this study, fraxinol strongly promoted melanin synthesis in B16-F10 cells. Our results suggest that fraxinol is a potent candidate for the development of pigmentation agents; however, the efficacy and safety of fraxinol-induced melanogenesis stimulation must be evaluated in vitiligo animal models and humans.

## 4. Materials and Methods

### 4.1. Chemicals and Reagents

Fraxinol (6-hydroxy-5,7-dimethoxychromen-2-one) was purchased from Cayman Chemical (Ann Arbor, MI, USA). Dulbecco’s modified Eagle medium (DMEM), fetal bovine serum, penicillin/streptomycin, and trypsin-ethylenediaminetetraacetic acid were purchased from HyClone (Logan, UT, USA), Corning, Inc. (Corning, NY, USA), Gibco (Grand Island, NY, USA), and Welgene (Gyeongsan-si, Republic of Korea), respectively. Dimethyl sulfoxide, α-MSH, NaOH, and MTT were obtained from Sigma–Aldrich (St. Louis, MO, USA). Antibodies against tyrosinase, TRP-1, TRP-2, MITF, and phosphorylated CREB were purchased from Santa Cruz Biotechnology (Dallas, TX, USA). H89 cells were purchased from Abcam (Cambridge, UK). Enhanced chemiluminescence kits were obtained from GE Healthcare (Buckinghamshire, UK).

### 4.2. Cell Culture

B16-F10 mouse melanoma cells were cultured in DMEM supplemented with 1% penicillin/streptomycin and 10% fetal bovine serum at 37 °C in a humidified atmosphere containing 5% CO_2_.

### 4.3. Cell Viability Assay

Cell viability was determined using an MTT assay. B16-F10 mouse melanoma cells (3 × 10^3^ cells/well) were seeded into 96-well plates and allowed to stabilize for 24 h. The cells were incubated with various concentrations of fraxinol (20, 40, 60, 80, or 100 μM) for 48 h and then treated with MTT (dissolved in DMEM to 5 mg/mL) for 2 h. The solution was removed, and dimethyl sulfoxide was added to dissolve the formazan crystals. The absorbance was measured at 570 nm using a Varioskan LUX microplate reader (Thermo Fisher Scientific, Waltham, MA, USA).

### 4.4. Measurement of Melanin Content

B16F10 mouse melanoma cells (2.5 × 10^4^ cells/well) were seeded into 6-well plates. After 24 h of incubation, fraxinol (20, 40, 60, 80, or 100 μM) was added, and the cells were incubated for 48 h. α-MSH (0.5 nM) was used as a positive control. The cell pellets were harvested and dissolved in 100 μL 1N NaOH at 60 °C for 2 h. Absorbance was measured at 405 nm using a Varioskan LUX microplate reader.

### 4.5. Intracellular Tyrosinase Activity and Cell-Free Mushroom Tyrosinase Activity

B16-F10 mouse melanoma cells (5 × 10^4^ cells/well) were seeded into 6-well plates. After 24 h of incubation, the cells were treated with fraxinol (20, 40, 60, 80, or 100 μM) for 48 h and then lysed with 0.5% Triton X-100 at 4 °C overnight. The cell lysates were vortexed and centrifuged at 13,000× *g* for 10 min. Next, 25 μL of each lysate and 100 μL of L-DOPA (1 mg/mL) were mixed and incubated at 37 °C for 1 h. Commercially available tyrosinase extracted from mushrooms was used in a cell-free system. L-DOPA (1 mg/mL), mushroom tyrosinase (250 U/mL), and fraxinol (0, 20, 40, 60, 80, or 100 μM) were incubated at 37 °C for 15 min. Absorbance was measured at 490 nm using a Varioskan LUX microplate reader (Thermo Fisher, Waltham, MA, USA).

### 4.6. L-DOPA Staining

B16-F10 cells were fixed in 2% paraformaldehyde in phosphate-buffered saline for 10 min and then incubated with 100% methanol for 10 min. After washing with phosphate-buffered saline, the cells were incubated in L-DOPA (1 mg/mL) for 6 h at 37 °C prior to observation under a light microscope.

### 4.7. RNA Isolation and qRT-PCR Analysis

Total RNA was isolated from cultured cells using TRIzol reagent (Invitrogen, Carlsbad, CA, USA). The RNA was quantified using a NanoDrop spectrophotometer (Thermo Fisher Scientific). Total RNA (2 μg) was reverse-transcribed using a SuperScript III cDNA synthesis kit (Invitrogen) according to the manufacturer’s instructions. qRT-PCR was performed using the StepOnePlus real-time PCR system (Applied Biosystems, Foster City, CA, USA) with PowerUp SYBR Green Master Mix (Applied Biosystems) and the indicated primers (Table 1). The threshold cycle (Ct) values of the genes of interest were normalized to that of β-actin.

### 4.8. Western Blot Assay

B16-F10 mouse melanoma cells (2.5 × 10^5^ cells) were seeded into 60-mm dishes and incubated with fraxinol. The cells were lysed in cold RIPA buffer containing protease inhibitors (1 mM of PMSF, 1 μg/mL of aprotinin, 1 μg/mL of pepstatin, and 1 μg/mL of leupeptin) and phosphatase inhibitors (30 mM of NaF and 1.5 mM of Na_3_VO_4_) using a sonicator, and then vortexed every 5 min for 30 min on ice. The cell lysate was centrifuged at 13,000× *g* for 10 min at 4 °C. The supernatants were collected, and the protein concentration was measured using the Bradford assay. After boiling in sample buffer, 10 μg of protein was separated on 12% sodium dodecyl sulfate polyacrylamide gels and then transferred onto nitrocellulose membranes (Cytiva, Marlborough, MA, USA). The membrane was blocked for 30 min with a 5% skim milk solution and incubated overnight with a primary antibody. Primary antibodies were diluted as follows: phosphorylated CREB (1:1000), MITF (1:10,000), TYR (1:500), TRP1 (1:5000), and TRP2 (1:1000). Equal loading was assessed using an anti-β-actin antibody. After washing every 5 for 30 min, the membranes were incubated with horseradish peroxidase-conjugated secondary antibodies at a dilution of 1:10,000 for 1 h. The targeted proteins were detected using enhanced chemiluminescence western blotting detection reagents and visualized using an iBright^TM^ CL1500 Imaging System (Applied Biosystems, Walthan, MA, USA).

### 4.9. Statistical Analysis

The results are expressed as the mean ± SD, and statistical analysis was performed using the *t*-test. A one-tailed value of *p* < 0.05, was considered to indicate a significant difference. All data were obtained from two or three independent experiments.

## Figures and Tables

**Figure 1 molecules-27-01549-f001:**
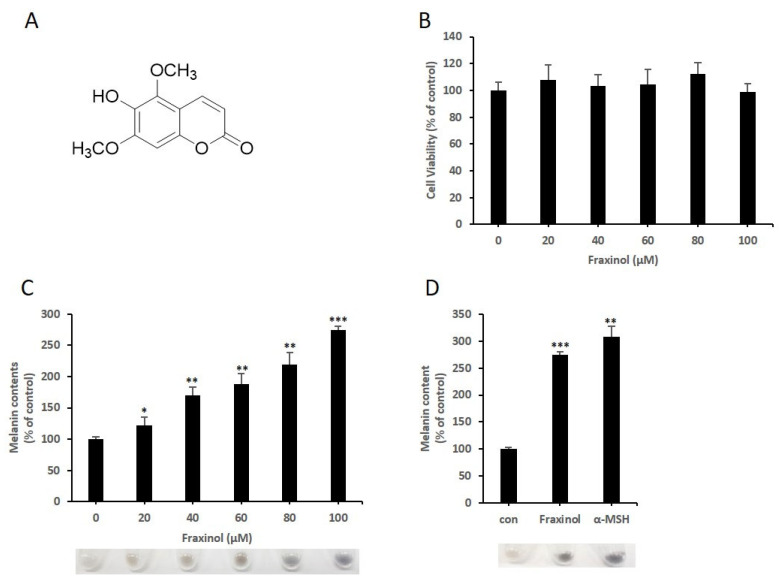
Effects of fraxinol on cell viability and melanin production in B16F10 cells. (**A**) Chemical structure of fraxinol. (**B**) Cytotoxicity of fraxinol in B16F10 cells. Cells were incubated with various concentration of fraxinol (0, 20, 40, 60, 80, or 100 μM) for 48 h. Cell viability was measured using the MTT assay. The data are shown as the means ± SD; *n* = 3. (**C**) Effect of fraxinol on melanin accumulation in B16-F10 cells. Cells were treated with fraxinol (0, 20, 40, 60, 80, or 100 μM) for 48 h. (**D**) Effect of fraxinol and α-MSH on melanin content in B16-F10 cells. Cells were treated with fraxinol (100 μM) or α-MSH (100 nM) for 48 h. Cellular melanin contents were measured as described in the Materials and Methods section and expressed as percentages compared to the respective value obtained for the control cells (non-treated cells). The data are presented as the means ± SD; *n* = 3. * *p* < 0.05, ** *p* < 0.01, *** *p* < 0.001 compared to 0 μM (no treatment).

**Figure 2 molecules-27-01549-f002:**
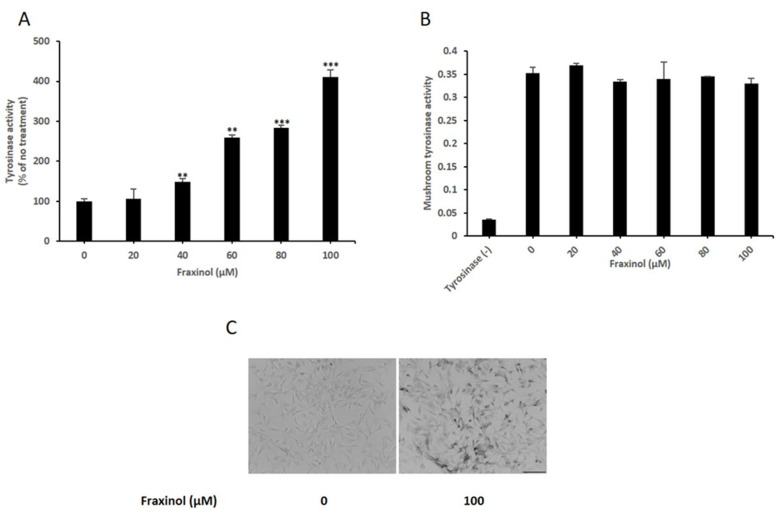
Effects of fraxinol on tyrosinase activity in B16F10 cells. (**A**) B16F10 cells were treated with fraxinol (0, 20, 40, 60, 80, or 100 μM) for 48 h. The tyrosinase activity is presented as percentages compared to the respective value obtained for the control cells (no treatment). (**B**) Effect of fraxinol on mushroom tyrosinase activity in a cell free system was determined as described in the Material and Methods section. (**C**) Intracellular tyrosinase activity was observed using a light microscope after DOPA staining (scale bar, 100 μm). The data are presented as the means ± SD; *n* = 3. ** *p* < 0.01, *** *p* < 0.001 compared to 0 μM (no treatment).

**Figure 3 molecules-27-01549-f003:**
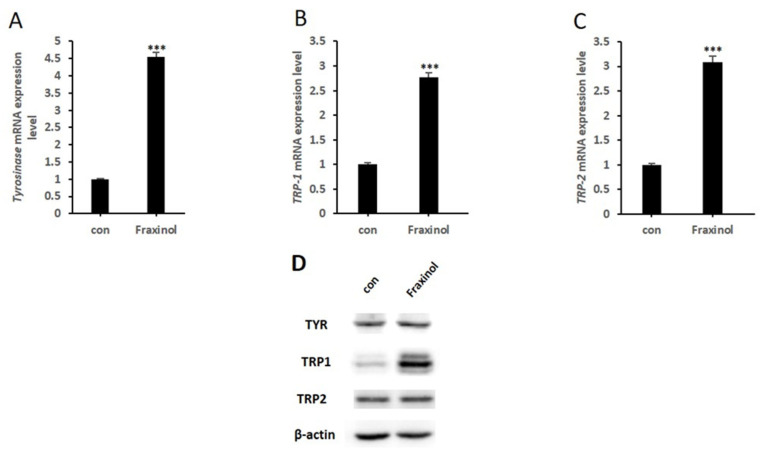
Effects of fraxinol on mRNA and protein expression of tyrosinase, TRP-1, and TRP-2. B16-F10 cells were treated with fraxinol (100 μM). (**A**) *TYR*, (**B**) *TRP-1*, and (**C**) *TRP-2* mRNA levels were determined using qRT-PCR. (**D**) Tyrosinase, TRP-1, and TRP-2 protein levels were determined using western blotting. The data are presented as the means ± SD; *n* = 2. *** *p* < 0.001 compared to the control (no treatment).

**Figure 4 molecules-27-01549-f004:**
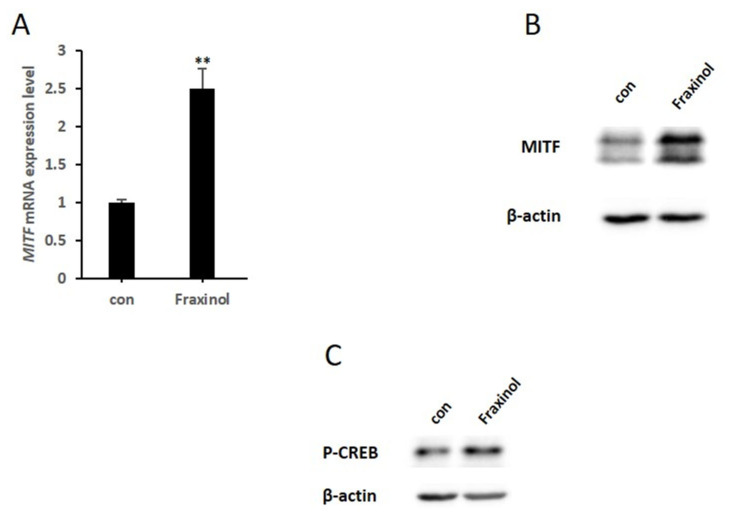
Effects of fraxinol on mRNA and protein expression of MITF and phosphorylation of CREB. B16F10 cells were treated with fraxinol (100 μM). (**A**) *MITF* mRNA expression was detected using qRT-PCR at 12 h after fraxinol treatment. (**B**) MITF protein expression was detected using western blotting at 24 h after fraxinol treatment. (**C**) Expression level of phosphorylated CREB was examined using western blotting at 4 h after fraxinol treatment. The data are presented as the means ± SD; *n* = 2. ** *p* < 0.01 compared to control (no treatment).

**Figure 5 molecules-27-01549-f005:**
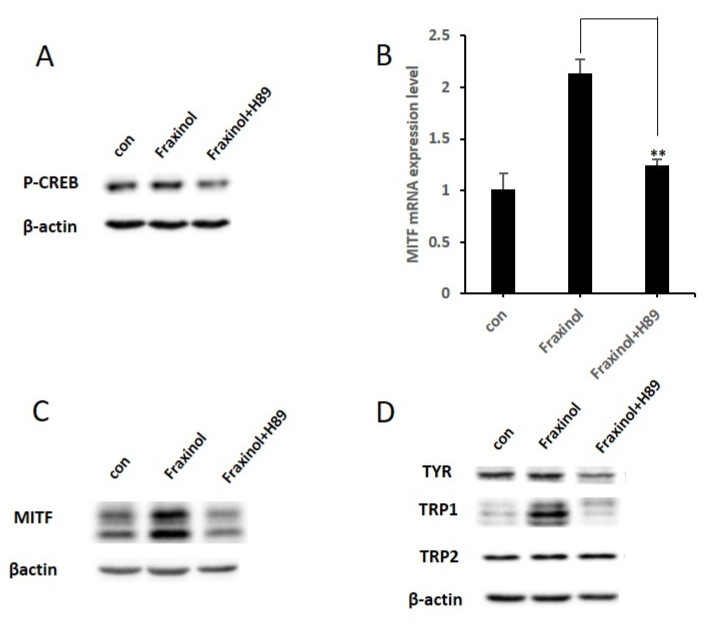
Effects of PKA inhibition on fraxinol-induced phosphorylation of CREB and expression of MITF mRNA and proteins in B16F10 cells. B16-F10 cells pretreated with or without H89 (10 μM) for 1 h were stimulated with fraxinol (100 μM). (**A**) Phosphorylation of CREB was detected at 4 h after fraxinol treatment using western blotting. (**B**) *MITF* mRNA expression was detected at 12 h after fraxinol treatment using qRT-PCR. (**C**) MITF protein expression was detected at 24 h after fraxinol treatment using western blot analysis. (**D**) Tyrosinase, TRP-1, and TRP2 protein expression was detected at 48 h after fraxinol treatment using western blotting. The data are presented as the means ± SD; *n* = 2. ** *p* < 0.01 compared to the fraxinol-treated group.

**Figure 6 molecules-27-01549-f006:**
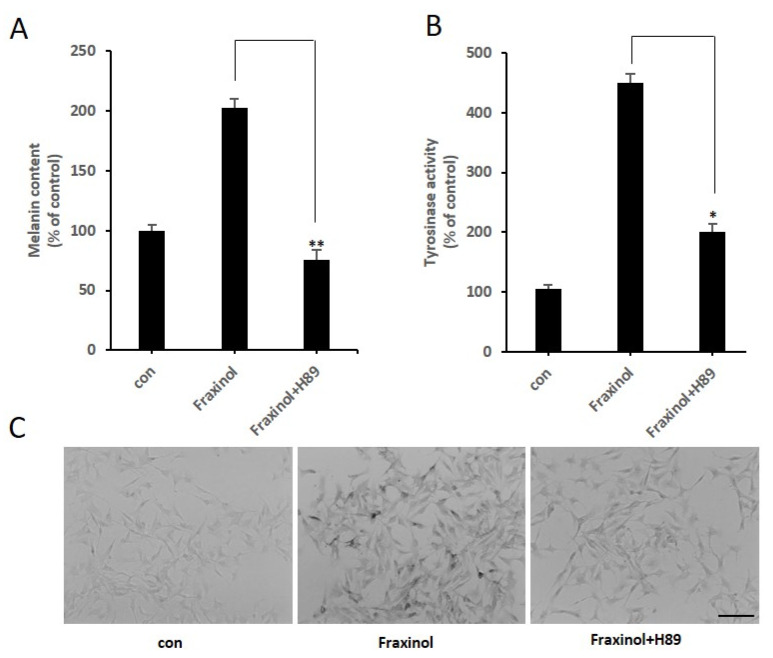
Effects of a PKA inhibitor on fraxinol-induced tyrosinase activation and melanin contents in B16F10 cells. B16F10 cells pretreated with or without H89 (10 μM) for 1 h were stimulated with fraxinol (100 μM) for 48 h. (**A**) Melanin content is expressed as percentages compared to the respective value obtained for the control cells (non-treated cells). (**B**) Tyrosinase activity is presented as a percentage compared to the respective value obtained for the control cells (no treatment). (**C**) Intracellular tyrosinase activity was observed using a light microscope after DOPA staining (scale bar, 100 μm). The data are presented as the means ± SD; *n* = 3. * *p* < 0.01, ** *p* < 0.001 compared to the fraxinol-treated group.

**Table 1 molecules-27-01549-t001:** Sequences of primers used in qRT-PCR.

Target		Sequence (5′ to 3′)	Accession No.
MITF	Forward	GAAACCTTGCTATGCTGGAAATG	BC108976.2
	Reverse	GAGCTTGCTGTATGTGGTACTT	
TYR	Forward	CCTCATGGACAAAGACGACTAC	BC079678.1
	Reverse	CCTTTCAGTCCCACTCTGTTT	
TRP1	Forward	GCTCCAGACAATCTGGGATATG	BC076598.1
	Reverse	AGTAACAACGCAGCCACTAC	
TRP2	Forward	CCTGTCTCTCCAGAAGTTTGAC	BC082330.1
	Reverse	CCAGTGTTCCGTCTGCTTTA	
β-actin	Forward	GAGGTATCCTGACCCTGAAGTA	NM_007393
	Reverse	CACACGCAGCTCATTGTAGA	

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
