# Peer review of "Fraxinol Stimulates Melanogenesis in B16F10 Mouse Melanoma Cells through CREB/MITF Signaling"

_molecules, 2022, doi:10.3390/molecules27051549_

Round 1
Reviewer 1 Report
This paper titled “Fraxinol Stimulates Melanogenesis in B16F10 Mouse Melanoma Cells Through CREB/MITF Signaling” is interesting. This manuscript could be considered for publication in Molecules after major revising.
My comments are as follow:
- Check the abbreviations, the text format of full text.
- English should be improved.
- The protocol of isolated and identified Fraxinol should be described in detail.
- Please compare your data with previous studies in the result and discussion section.
Author Response
Thank you very much for giving us an opportunity to submit a revised manuscript. Our responses to the reviewers’ comments are included in this letter. In addition, we have modified some parts of the manuscript in response to the reviewers’ comments. Modified parts are indicated in yellow. We hope that you find the revised manuscript acceptable for publication in ‘Molecules’
Comments and Suggestions for Authors
This paper titled “Fraxinol stimulated Melanogenesis in B16F10 Mouse Melanoma Cells Through CREB/MITF Signaling” is interesting. This manuscript could be considered for publication in Molecules after major revising.
My comments are as follow:
- Check the abbreviations, the text format of full text.
We have checked all abbreviations in the full text and made the following corrections.
Page 2, lines 59–60: Microphthalmia-associated transcription factor (MITF)
Page 2, lines 65–66: cyclic adenosine monophosphate (cAMP)/cAMP–dependent protein kinase A (PKA)
Page 2, line 67: cAMP response element-binding protein (CREB)
Page 2, line 91: 3-(4,5-dimethylthiazol-2-yl)-2,5-diphenyltetrazolium bromide (MTT)
Page 3, line 115: We replaced tyrosinase, tyrosinase-related protein-1, and tyrosinase-related protein-2 with abbreviations (TYR, TRP1, and TRP2)
Once an abbreviation has been defined, it is used consistently thereafter.
- English should be improved
According to the reviewer’s recommendation, the text has been revised by a native English speaker employed by “Editage,” a professional English editing company. The edited parts are indicated with red letter. The “certificate of English editing” is attached at the end of the cover letter for the editor.
- The protocol of isolated and identified Fraxinol should be described in detail
As described in 4.1 Chemicals and Reagents, fraxinol used in the study was purchased from Cayman Chemical (Ann Arbor, MI, USA). However, we have added the following paper as a reference (line 81): (Reference 24)
Akter et al. isolated fraxinol from Fraxinus mandshurica bark, which was published in Natural Product Sciences 26(2): 158‒164 (2020).
- Please compare your data with previous studies in the result and discussion section.
According to the reviewer’s recommendation, we have added the data (Figure 1D), including the melanin content in B16-F10 cells treated with alpha-melanocyte-stimulating hormone (α-MSH). Previous studies confirmed that α-MSH stimulates melanogenesis in melanocytes by activating melanocortin 1 receptor. Therefore, we compared the α-MSH-induced melanin content with that induced by fraxinol. In addition, in the Discussion section, we described the melanogenesis-inducing effects of fraxinol compared with those of other compounds (ginsenosides and glycyrrhizin), which have been reported to stimulate melanogenesis in melanocytes.
Page 2-3, lines 96-99: We also compared the fraxinol-induced melanin content with the alpha-melanocyte stimulating hormone (α-MSH)-stimulated melanin content (Figure 1D). Compared to untreated cells, fraxinol and α-MSH increased melanin levels by approximately 2.5- and 3-fold, respectively.
Page 8-9, lines 223-230: We used αMSH (100 nM) as a positive control to induce melanin production. Previous studies confirmed that α-MSH stimulates melanogenesis in melanocytes by activating the melanocortin 1 receptor, with the melanin-inducing effect saturated at approximately 100 nM [35-38]. Fraxinol and α-MSH increased the levels of intracellular melanin by 2.5- and 3-fold compared to that in non-treated cells. Plants or active small molecules of natural origin have been reported to stimulate melanogenesis. Among the active compounds, ginsenoside and glycyrrhizin showed the strongest melanogenesis-inducing effects at concentrations of 100–300 μM and 1 mM, respectively, compared to fraxinol [39,40].
Page 9, lines 253-256: Ginsenosides, also, stimulate melanogenesis via the PKA/CREB/MITF pathway [39] and glycyrrhizin induces melanogenesis by blocking glycogen synthase kinase 3β phosphorylation and by enhancing CREB phosphorylation [40].

Reviewer 2 Report
The manuscript “Fraxinol Stimulates Melanogenesis in B16F10 Mouse Melanoma Cells Through CREB/MITF Signaling” by Moon et al. presents interesting data on effect of fraxinol on production of melanin in B16F10 melanoma cells. The authors showed that fraxinol potentiates the production of melanin by activation of tyrosinase and proposed that fraxinol activates tyrosinase through PKA-mediated activation of CREB/MITF pathway. Overall, the manuscript is well written and the results are presented clearly. I have only a few suggestions and recommendations that need to be followed before accepting the manuscript.
Minor Comments and Suggestions:
Page 3, lane 95; Figure 1A,B,C: Authors should specify in the Legends of Figures the number of biological or technical replicates that were averaged to get mean values. On page 10/lane 316 they only generally stated that “All data were obtained from two or three independent experiments”. Additionally, data reflecting the intra- and extracellular melanin contents in B16F10 cells treated with alpha-melanocyte stimulating hormone (α-MSH) could be added in Figure 1 and compared with the effect of fraxinol.
Page 4, lane 124-125; Figure 2B: There are no details on a “Cell free mushroom tyrosinase activity assay” in the Material and Methods section.
Page 5, lanes 153-154: This sentence can be simplified as: This result suggests that fraxinol-induced increase in melanogenic enzymes can be attributed to the increase in MITF expression.”
Page 5, lanes 156-157: Authors stated that “As presented in Figure 4C, the expression levels of phosphorylated CREB (P-CREB) were enhanced by fraxinol treatment”. Did they test if phosphorylation of CREB persist even after 4 h? If possible, the authors should test or discuss if there is a time dependent effect of fraxinol treatment on phosphorylation of CREB and for how long is CREB phosphorylated in fraxinol treated cells.
Figure 5 and Figure 6: F+H89 should be replaced with Fraxinol+H89.
Figure 6: Why the values of melanin contents or data on tyrosinase activity presented in Figure 6 differ from those shown in Figure 1C and Figure 2A?
Page 9, lane 250: Chemical name of fraxinol, 6-hydroxy-5,7-dimethoxychromen-2-one, should be added in the Material and Methods section.
Page 10, lane 307: What was dilution of primary antibodies? Please, specify.
Author Response
Thank you very much for giving us an opportunity to submit a revised manuscript. Our responses to the reviewers’ comments are included in this letter. In addition, we have modified some parts of the manuscript in response to the reviewers’ comments. Modified parts are indicated in yellow. We hope that you find the revised manuscript acceptable for publication in ‘Molecules’
Comments and Suggestions for Authors
The manuscript “Fraxinol Stimulates Melanogenesis in B16F10 Mouse Melanoma Cells Through CREB/MITF Signaling” by Moon et al. presents interesting data on effect of fraxinol on production of melanin in B16F10 melanoma cells. The authors showed that fraxinol potentiates the production of melanin by activation of tyrosinase and proposed that fraxinol activates tyrosinase through PKA-mediated activation of CREB/MITF pathway. Overall, the manuscript is well written and the results are presented clearly. I have only a few suggestions and recommendations that need to be followed before accepting the manuscript
Thank you very much for your positive response.
Minor Comments and Suggestions:
- Page 3, lane 95; Figure 1A,B,C: Authors should specify in the Legends of Figures the number of biological or technical replicates that were averaged to get mean values. On page 10/lane 316 they only generally stated that “All data were obtained from two or three independent experiments”.
In response to the reviewer’s comment, we have added the number of biological replicates (n = 2 or 3) used to obtain the mean values in the legends of Figure 1–6.
- Additionally, data reflecting the intra- and extracellular melanin contents in B16F10 cells treated with alpha-melanocyte stimulating hormone (α-MSH) could be added in Figure 1 and compared with the effect of fraxinol
In accordance with the reviewer’s suggestion, we have added Figure 1D to show the melanin content in B16-F10 cells treated with α-melanocyte-stimulating hormone (α-MSH). In addition, we have described the comparison of melanin content induced by α-MSH and effects of fraxinol in the Result and Discussion section.
Page 2-3, lines 96-99: We also compared the fraxinol-induced melanin content with the alpha-melanocyte stimulating hormone (α-MSH)-stimulated melanin content (Figure 1D). Compared to untreated cells, fraxinol and α-MSH increased melanin levels by approximately 2.5- and 3-fold, respectively.
Page 8, lines 223-227: We used αMSH (100 nM) as a positive control to induce melanin production. Previous studies confirmed that α-MSH stimulates melanogenesis in melanocytes by activating the melanocortin 1 receptor, with the melanin-inducing effect saturated at approximately 100 nM [35-38]. Fraxinol and α-MSH increased the levels of cellular melanin by 2.5- and 3-fold compared to that in non-treated cells.
- Page 4, lane 124-125; Figure 2B: There are no details on a “Cell free mushroom tyrosinase activity assay” in the Material and Methods section.
Based on this recommendation, we added cell-free mushroom tyrosinase activity to the Intracellular Tyrosinase Activity section in the Materials and Methods and replaced Intracellular Tyrosinase Activity with Intracellular Tyrosinase Activity and Cell-free Mushroom Tyrosinase Activity.
Page 10, lines 301-304: Commercially available tyrosinase extracted from mushrooms was used in a cell-free system. L-DOPA (1 mg/mL), mushroom tyrosinase (250 U/mL), and fraxinol (0, 20, 40, 60, 80, or 100 μM) were incubated at 37 °C for 15 min.
- Page 5, lanes 153-154: This sentence can be simplified as: This result suggests that fraxinol-induced increase in melanogenic enzymes can be attributed to the increase in MITF expression.”
We have simplified the sentence based on your suggestion (page 5, lines 159-161).
- Page 5, lanes 156-157: Authors stated that “As presented in Figure 4C, the expression levels of phosphorylated CREB (P-CREB) were enhanced by fraxinol treatment”. Did they test if phosphorylation of CREB persist even after 4 h? If possible, the authors should test or discuss if there is a time dependent effect of fraxinol treatment on phosphorylation of CREB and for how long is CREB phosphorylated in fraxinol treated cells.
We checked the level of phosphorylated CREB at 24 h after fraxinol treatment and found that it was still increased at this time point. Based on the reviewer’s suggestion, we have mentioned this result as “data not shown” and described this information in the Discussion section.
Please see the western blotting data below.
Page 9, lines 246-249: The phosphorylation of CREB was enhanced at 4 h after fraxinol treatment and was still increased at 24 h (data not shown). This result indicates that increased phosphorylated CREB contributes to the enhancement of melanin production until 24 h after treatment with fraxinol.
- Figure 5 and Figure 6: F+H89 should be replaced with Fraxinol+H89.
According to this recommendation, we have replaced F+H89 with Fraxinol+H89 (Figure 5 and 6).
- Figure 6: Why the values of melanin contents or data on tyrosinase activity presented in Figure 6 differ from those shown in Figure 1C and Figure 2A?
We obtained data from several independent experiments and one representative dataset. The cell passage used in this study ranged from three to ten in each independent experiment. Although the melanin content and tyrosinase activity showed increasing trends following exposure to fraxinol, the values for the melanin content and tyrosinase activity somewhat differed in each experiment depending on the cell passage number or cell conditions. In response to the reviewer’s question, we tried to find the data showing a similar value as in Figure 1C and 2A among our multiple datasets and replaced them to Figure 6A and 6B.
- Page 9, lane 250: Chemical name of fraxinol, 6-hydroxy-5,7-dimethoxychromen-2-one, should be added in the Material and Methods section.
Based on this suggestion, we have added 6-hydroxy-5,7-dimethoxychromen-2-one to the Chemical and Reagents section in the Materials and Methods. (page 9, line 268)
- Page 10, lane 307: What was dilution of primary antibodies? Please, specify
Based on this suggestion, we have added the fold-dilution of each primary antibody used in the western blot assay to the Materials and Methods section.
Page 11, lines 330-332: Primary antibodies were diluted as follows: phosphorylated CREB (1:1,000), MITF (1:10,000), TYR (1:500), TRP1 (1:5,000), and TRP2 (1:1,000).

Round 2
Reviewer 1 Report
English should be checked by a native speaker.